

# Boundary conditions for extremal black holes from 2d gravity

Stephane Detournay[1★], Thomas Smoes[1†] and Raphaela Wutte[1,2‡]

**1** Physique Mathématique des Interactions Fondamentales, Université Libre de Bruxelles,
Campus Plaine - CP 231, 1050 Bruxelles, Belgium
**2** Department of Physics and Beyond: Center for Fundamental Concepts in Science,
Arizona State University, Tempe, Arizona 85287, USA

★ sdetourn@ulb.ac.be , † thomas.smoes@ulb.be , ‡ rwutte@hep.itp.tuwien.ac.at

## Abstract

We devise new boundary conditions for the near-horizon geometries of extremal BTZ and
Kerr black holes, as well as for the ultra-cold limit of the Kerr-de Sitter black hole. These
boundary conditions are obtained as the higher-dimensional uplift of recently proposed
boundary conditions in two-dimensional gravity. Their asymptotic symmetries consist in
the semi-direct product of a Virasoro and a current algebra, of which we determine the
central extensions.

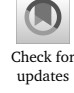

# 1 Introduction and Outlook

When formulating a physical problem, the equations of motion have to be supplemented by boundary conditions (BCs) on the dynamical variables. In fact, the latter turn out to be as important as the former [1] (cited in [2]). This is especially clear when the theory is formulated in terms of an action principle and the partition function defined through a path integral: the boundary conditions specify the off-shell configurations over which the integral has to be performed. Systems with identical field equations but different boundary conditions could describe significantly distinct physical phenomena and exhibit different contents (e.g. closed/open strings, Dirichlet vs Neumann BCs).

Boundary conditions play a crucial role in gauge theories, in particular in theories of gravity. There, the set of metrics satisfying given equations of motion and boundary conditions constitute the configuration space of the theory, which can be identified with its phase space. The identification of the symmetries of the phase space are of crucial importance since one expects, upon quantization, that the Hilbert space of the corresponding quantum theory will fall into a representation of the symmetry group, for instance in the spirit of the geometric quantization program [3, 4].

In gauge theories, the symmetries of the phase space, mapping one solution onto another with distinct physical charges, are of great importance. These are called *asymptotic symmetries* and form the *asymptotic symmetry group* (ASG). The study of asymptotic symmetries in gravity theories has a long history that started in 1962 with the founding papers [5, 6] which identified the BMS group of supertranslations and Lorentz transformations as ASG of four-dimensional asymptotically flat spacetimes. It was later extended to include superrotations in [7–9] and diffeomorphisms on the 2-sphere in [10, 11]. The renewed interest in BMS symmetries is largely due to recent work on BMS invariance of scattering amplitudes [12] and the "infrared triangle" relating BMS supertranslation symmetries, Weinberg's soft graviton theorem and the displacement memory effect [13].

Equally impactful is the discovery by Brown and Henneaux of two-dimensional conformal symmetry in the asymptotic structure of $AdS_3$ gravity [14], an early precursor of the AdS/CFT correspondence [15]. It brought deep insights into the holographic nature of gravity and in particular the identification of microscopic degrees of freedom for specific classes of black holes, either asymptotically $AdS_3$ (the BTZ black hole [16, 17]) [18] or with an $AdS_3$ factor in their near-horizon geometry [19]. The three-dimensional situation in flat space has been addressed more recently, identifying the $BMS_3$ asymptotic symmetry algebra at null infinity [20, 21] and at spatial infinity [22]. The flat limit from $AdS_3$ to Minkowski was described in [21] for the symmetry algebra, and for the full phase space in [23]. The flat spacetime cosmologies [24, 25] – the flat counterparts of the BTZ black holes – and their thermodynamical interpretation in terms of $BMS_3$ symmetries were addressed in [26, 27]. Interestingly, the non-uniqueness of the ASG given a vacuum solution and non-trivial zero-mode solutions has been brought to light only rather recently. Superrotations in four-dimensional asymptotically flat space have been introduced almost half a century after the works of Bondi, van der Burg, Metzner and Sachs. In three-dimensional gravity, a variety of alternative boundary conditions – allowing e.g. for a fluctuating boundary metric, in contrast with the Dirichlet-like Brown-Henneaux boundary conditions – have been proposed in recent years both for $AdS_3$ [28–33] and Minkowski space [34–36] exhibiting in general different ASGs, hence potentially different field theory dual interpretations. A particular way of relating different ASGs in three dimensions has been discussed in [37].

Among holographic dualities involving AdS spaces, the two-dimensional case has always stood out as more challenging. The boundary of $AdS_2$ consists in two disconnected pieces, and finite energy excitations have been observed to destroy the asymptotic geometry [38, 39]. This

has long been a hindrance for a microscopic understanding of extremal higher-dimensional black holes, as these generally exhibit a near-horizon geometry including an AdS$_2$ factor [40, 41] when the cosmological constant is non-positive (we will later discuss a situation where AdS$_2$ in replaced by Mink$_2$ for the near-horizon limit of the ultra-cold Kerr-de Sitter black hole [42]). It has however recently been found how to circumvent these obstructions and identify the relevant degrees of freedom describing the low energy physics driving a black hole away from extremality. It consists in considering *nearly*-AdS$_2$ holography by including the leading corrections away from pure AdS$_2$ [43, 44] (for reviews, see e.g. [45, 46] or App.B of [47]). The physics near the horizon of near-extremal black holes in higher dimensions can be shown to be universally described by a particular occurrence of two-dimensional dilaton gravity theory – JT gravity [48, 49], with certain Dirichlet boundary conditions at the boundary of AdS$_2$. The latter exhibit time-reparametrization invariance whose generators[1] are reminiscent of (one half of) the Brown-Henneaux ones [50–52]. Again, like in higher dimensions, different sets of boundary conditions with different symmetries can be considered [53]. Recently, new boundary conditions for AdS$_2$ have been proposed [54], where the usual time-reparametrization symmetry is enhanced with an additional local U(1) symmetry, extending the symmetry algebra to a Virasoro-Kac-Moody $U(1)$ algebra. The latter represent the symmetries of a so-called Warped CFT (WCFT) [55, 56], a two-dimensional non-relativistic theory with chiral scale invariance and $SL(2, R) \times U(1)$ global symmetry (see [29, 57–62] for some of their properties).

The goal of the present work will be to explore new boundary conditions for extremal black holes, in particular determine whether the boundary conditions of [54] and [63] can be uplifted to the near-horizon geometry in higher dimensions. Our work can thus be regarded as a proof of principle that certain boundary conditions existing in 2d gravity have a natural uplift to higher dimensions.

Motivations stem from the ubiquity of AdS$_2$ in the near-horizon geometry of extremal black holes, but also from the Kerr/CFT correspondence [64] – an attempt to relate four-dimensional extremal Kerr black holes to a chiral CFT in two dimensions. The argument there parallels the connection between AdS$_3$ and 2d CFTs, where the AdS$_3$ near-region throat geometry is replaced with the NHEK (near-horizon extreme Kerr) geometry found by Bardeen and Horowitz [65] via a near-horizon limit. Constant polar sections of the NHEK geometry consist in deformations of AdS$_3$, termed Warped AdS$_3$ (WAdS$_3$) spaces [66–72], where the original undeformed $SO(2, 2)$ isometries get broken down to $SL(2, R) \times U(1)$. Holographic properties of WAdS$_3$ spaces have been explored over the years [56, 73–89] as a toy model for Kerr black holes. For generic Kerr black holes, the relevance of WCFTs was pointed out in [90] in the spirit of [91]. In the extremal limit, the question is still open.

The Kerr/CFT proposal is based on boundary conditions extending the $U(1)$ part of the isometry group into a Virasoro algebra, whose computed central charge allowed to reproduce the macroscopic Bekenstein-Hawking extremal Kerr entropy. This was one of the landmarks of the original proposal.[2] From a gravity perspective these boundary conditions might seem unnatural, as their symmetries do not include all the exact symmetries of the background. Soon after the Kerr/CFT proposal, other boundary conditions have been proposed extending instead the $SL(2, R)$ part of the isometries, but found vanishing central extensions [94, 95]. In this work, we will propose new boundary conditions for the NHEK geometry, inspired by the Godet-Marteau analysis in two dimensions [54]. One feature of these boundary conditions and their symmetries is the dependence of the generators on (retarded) time. Extracting a non-

---

[1]Note that the reparameterisation symmetry is broken both spontaneously by pure AdS$_2$ and explicitly due to the non-trivial boundary condition for the dilaton.

[2]This is currently being debated in recent works suggesting instead a vanishing entropy at low temperatures [92, 93].

trivial symmetry algebra therefore requires to integrate charges over time instead of the usual constant-time, angular integration. This procedure has been applied both in two and higher dimensions [51, 53, 96, 97]. Integration over time produces time-averaged charges which can be seen to give a canonical representation of the asymptotic symmetry algebra with non-trivial central extensions. The procedure can also be interpreted from the boundary perspective, in particular when the putative dual theory is two-dimensional (CFT, WCFT, or other) and enjoys modular invariance. A modular invariant field theory at finite chemical potentials is naturally defined on a torus with two cycles, the spatial one (angular identifications) and the thermal one (in particular, time has a period set by the inverse temperature). Its partition function can be expressed either as a trace over states defined on spatial cycles (with charges integrated over a spatial cycle) and evolved with the usual hamiltonian operator, or as states defined on thermal cycles (hence with time periodic in particular and charges integrated over a thermal cycle) and evolved with the angular momentum operator. This yields one possible boundary interpretation of a bulk time integration and it is this interpretation that we will employ throughout this work.

The paper is organized as follows. As a warm-up, we devise in Sect. 2 new boundary conditions for the near-horizon limit of extremal BTZ black holes, the so-called selfdual orbifold. Kerr/CFT-like boundary conditions had appeared e.g. in [98]. Here we define a new phase space with WCFT symmetries of which we identify the non-trivial central extensions, the Virasoro one coinciding with the Brown-Henneaux central charge. In Sect. 3 we turn to boundary conditions including the NHEK geometry. Following a similar strategy, we define a phase space, identify their asymptotic symmetries, and compute the asymptotic charges. The latter are shown to satisfy through their Poisson bracket a WCFT algebra with non trivial central extensions both for the Virasoro and current algebra. The Virasoro central extensions is seen to match that of the original Kerr/CFT correspondence. We address a slightly different case in Sect. 4. It consists in boundary conditions including the near-horizon limit of the ultra-cold Kerr-de Sitter black hole in 4 dimensions (where the 3 horizons come to coincide). There is no known way to associate a CFT or any other boundary theory for that matter to the ultracold limit [42] (see however [99] studying the response of ultracold black holes to small perturbations). The latter does not fall in the general category of $AdS_2$ near-horizon geometry. Instead, the $AdS_2$ factor is replaced by two-dimensional Minkowski space. As it turns out, boundary conditions for $Mink_2$ have been proposed and their asymptotic symmetries determined [63, 100]. We uplift these boundary conditions to 4 dimensions, demonstrating that they yield well defined charges and asymptotic symmetry algebras, again consisting in a WCFT algebra of which we compute the central extensions. This provides a first step towards building a holographic dual for ultracold Kerr-dS black holes. In section 5 we conclude with a summary and interpretation of our results in the context of holography.

## 2 Extremal BTZ

### 2.1 Geometry and near-horizon Limit

The metric of the extremal BTZ black hole is

$$ds^2 = -\frac{(r^2 - r_h^2)^2}{r^2} dt^2 + \frac{r^2}{(r^2 - r_h^2)^2} dr^2 + r^2 \left( d\phi - \frac{r_h^2}{r^2} dt \right)^2, \tag{1}$$

where $r_h$ is the horizon radius and where the AdS radius $l$ has been set to one. We consider the change of coordinates

$$t = \frac{\tau}{\epsilon}, \quad r^2 = r_h^2 + \epsilon \rho, \quad \phi = \varphi + \frac{\tau}{\epsilon}, \tag{2}$$

and then study the near-horizon limit (NHL) by taking $\epsilon \to 0$. The extremal BTZ metric becomes [101]

$$ds^2 = \frac{1}{4}\frac{d\rho^2}{\rho^2} + 2\rho\, d\tau\, d\varphi + r_h^2 d\varphi^2$$
$$= \frac{1}{4}\frac{d\rho^2}{\rho^2} - \frac{\rho^2}{r_h^2}d\tau^2 + r_h^2\left(d\varphi + \frac{\rho}{r_h^2}d\tau\right)^2. \tag{3}$$

In order to apply Godet-Marteau boundary conditions on this metric, we will write it in a system of coordinates that is similar to the Bondi gauge described in [54] for AdS$_2$. We thus define new coordinates $(u, \hat{r}, \hat{\varphi})$ such that

$$\tau = \frac{u}{2} - \frac{1}{2\hat{r}}, \quad \rho = r_h\hat{r}, \quad \varphi = \frac{1}{2r_h}(\hat{\varphi} - \ln\hat{r}), \tag{4}$$

and the metric becomes

$$ds^2 = \frac{1}{4}(-\hat{r}^2 du^2 - 2du\, d\hat{r}) + \frac{1}{4}(\hat{r}\, du + d\hat{\varphi})^2. \tag{5}$$

From now on, we will omit "^" of the coordinates, keeping in mind that the new coordinates are different from the ones in (3).

## 2.2 Phase space and asymptotic Killing vectors

Inspired by the Godet-Marteau boundary conditions for AdS$_2$ [54], we consider the following family of metrics

$$ds^2 = \frac{1}{4}\left((-r^2 + 2P(u)r + 2T(u))du^2 - 2du\, dr\right) + \frac{1}{4}(r\, du + d\varphi)^2 \tag{6a}$$
$$= ds_{2d}^2 + \frac{1}{4}(r\, du + d\varphi)^2, \tag{6b}$$

where $P$ and $T$ are arbitrary functions of $u$. Here, the first part of the metric $ds_{2d}^2$ corresponds to boundary conditions that were previously imposed for 2d gravity [54]. The boundary conditions (6) can be obtained from (5) by applying the finite coordinate transformation

$$u \to \mathcal{F}(u), \quad r \to \frac{1}{\mathcal{F}'}\left(r + \mathcal{G}'(u)\right), \quad \varphi \to \varphi - \mathcal{G}(u). \tag{7}$$

The functions $P, T, \mathcal{F}$ and $\mathcal{G}$ are related by

$$P(u) = -\mathcal{G}'(u) + \frac{\mathcal{F}''(u)}{\mathcal{F}'(u)}, \tag{8}$$

$$T(u) = -\frac{1}{2}\mathcal{G}'(u)^2 + \mathcal{G}'(u)\frac{\mathcal{F}''(u)}{\mathcal{F}'(u)} - \mathcal{G}''(u). \tag{9}$$

The asymptotic Killing vectors generating the transformations (7) are given by

$$\xi = \epsilon(u)\partial_u + (-r\epsilon'(u) - \zeta'(u))\partial_r + \zeta(u)\partial_\varphi, \tag{10}$$

where $\epsilon(u)$ and $\zeta(u)$ are two arbitrary functions of $u$. By applying the Lie derivative on the metric (6), we can also find the variations of $P(u)$ and $T(u)$

$$\delta_\xi P = \epsilon P' + \epsilon' P + \epsilon'' + \zeta', \tag{11}$$

$$\delta_\xi T = \epsilon T' + 2\epsilon' T - \zeta' P + \zeta''. \tag{12}$$

Alternatively, we can define a perturbation $h_{\mu\nu}$ on the background metric (5) such that

$$h_{uu} = \mathcal{O}(r), \qquad h_{ur} = \mathcal{O}(r^{-2}), \quad h_{u\varphi} = \mathcal{O}(r^{-1}), \tag{13a}$$

$$h_{rr} = \mathcal{O}(r^{-3}), \quad h_{r\varphi} = \mathcal{O}(r^{-2}), \quad h_{\varphi\varphi} = \mathcal{O}(r^{-1}), \tag{13b}$$

and the vectors solving the asymptotic Killing equation are given by

$$\xi = \left(\epsilon(u) + \mathcal{O}(r^{-2})\right)\partial_u + \left(-r\epsilon'(u) - \zeta'(u) + \mathcal{O}(r^{-1})\right)\partial_r + \left(\zeta(u) + \mathcal{O}(r^{-2})\right)\partial_\varphi. \tag{14}$$

Fixing the coordinate system, by setting $g_{ur} = -1/4$, $g_{rr} = 0$ and $g_{r\varphi} = 0$, and assuming that the remaining components admit an expansion in powers of $r$

$$g_{uu} = r g_{uu1}(u,\varphi) + g_{uu0}(u,\varphi) + \mathcal{O}(r^{-1}), \tag{15a}$$

$$g_{u\varphi} = \frac{r}{4} + \frac{g_{u\varphi0}(u,\varphi)}{r} + \mathcal{O}(r^{-2}), \tag{15b}$$

$$g_{\varphi\varphi} = \frac{1}{4} + \frac{g_{\varphi\varphi-1}(u,\varphi)}{r} + \mathcal{O}(r^{-2}), \tag{15c}$$

one readily obtains that (6) is the unique class of metrics that solves the vacuum Einstein equations with a negative cosmological constant and the fall-off conditions (13). It is in this sense, that (13) and (14) are equivalent to (6) and (10). In the following, we will always work with a class of metrics instead of directly working with boundary conditions.

From now on, we assume that $u$ is periodic with period $L \in i\mathbb{R}$, where $i$ is the imaginary unit, and define the modes of the vectors (10) as

$$l_n = \xi\left(\epsilon = \frac{L}{2\pi}e^{2\pi i n u/L}, \ \zeta = 0\right), \quad j_n = \xi\left(\epsilon = 0, \ \zeta = \frac{L}{2\pi i}e^{2\pi i n u/L}\right), \tag{16}$$

where $n \in \mathbb{Z}$. The motivation for this arises from the Euclidean where Euclidean time is periodic with period $\beta$. Wick rotating, the period of Lorentzian time becomes $L = i\beta$ where $\beta$ is the temperature. These modes satisfy a warped Witt algebra under the Lie bracket:

$$i[l_m, l_n] = (m-n)l_{m+n}, \tag{17a}$$

$$i[l_m, j_n] = -n j_{m+n}, \tag{17b}$$

$$i[j_m, j_n] = 0. \tag{17c}$$

## 2.3 Charge algebra

The infinitesimal charge difference between two geometries $g_{\mu\nu}$ and $g_{\mu\nu} + h_{\mu\nu}$, where $h_{\mu\nu}$ is an infinitesimal perturbation, is given by

$$\delta Q_\xi[h, g] = \int_{\partial\Sigma} k_\xi[h, g]. \tag{18}$$

The differential form $k_\xi$ associated to an asymptotic Killing vector $\xi$ is defined by[3]

$$k_\xi[h, g] = \frac{\sqrt{-g}}{8\pi G}(d^{n-2}x)_{\mu\nu}\left(\xi^\mu \nabla_\sigma h^{\nu\sigma} - \xi^\mu \nabla^\nu h + \xi_\sigma \nabla^\nu h^{\mu\sigma} + \frac{1}{2}h\nabla^\nu \xi^\mu - h^{\rho\nu}\nabla_\rho \xi^\mu\right), \tag{19}$$

where $n$ is the space-time dimension, $\nabla$ is the covariant derivative of $g_{\mu\nu}$ and $h = g^{\mu\nu}h_{\mu\nu}$. One readily checks that integrating (18) along the direction of $\varphi$ over a constant $u$ surface and taking the limit $r \to \infty$, yields zero – all surface charges vanish. One may obtain non-zero

---

[3]See e.g. [102] for a pedagogical account and references.

surface charges by integrating (18) along the direction of $u$ over a constant $\varphi$ surface and then taking the limit $r \to \infty$, which is what we will do in what follows. The motivation for this stems from holography. In particular, as detailed in the introduction, assuming that the putative dual field theory is modular invariant, we can use this modular invariance to switch the angular and temporal cycle for the computation of the charges.

For this, we begin by defining the variation of the metric (6) as

$$h_{\mu\nu} \equiv \delta g_{\mu\nu} = \frac{\partial g_{\mu\nu}}{\partial P}\delta P + \frac{\partial g_{\mu\nu}}{\partial T}\delta T. \tag{20}$$

Computing the variation of the charges, we find

$$\delta Q_\xi = \frac{1}{16\pi G}\int_0^L du(\epsilon\delta T - \zeta\delta P). \tag{21}$$

We see that this expression can be directly integrated in order to obtain the finite charges

$$Q_\xi = \frac{1}{16\pi G}\int_0^L du(T(u)\epsilon(u) - P(u)\zeta(u)), \tag{22}$$

where the metric of the extremal black hole in the NHL, which has $P(u) = T(u) = 0$, has been chosen as the background metric. In particular, we define

$$L_n = Q_{l_n} = \frac{1}{16\pi G}\int_0^L du\, T(u)\frac{L}{2\pi}e^{2\pi i n u/L}, \tag{23}$$

$$J_n = Q_{j_n} = -\frac{1}{16\pi G}\int_0^L du\, P(u)\frac{L}{2\pi i}e^{2\pi i n u/L}. \tag{24}$$

Computing the algebra of these charges under the Dirac bracket yields

$$i\{L_m, L_n\} = i\delta_{l_n}L_m = (m-n)L_{m+n}, \tag{25a}$$

$$i\{L_m, J_n\} = i\delta_{j_n}L_m = -nJ_{m+n} - \frac{L}{16\pi G}m^2\delta_{m+n,0}, \tag{25b}$$

$$i\{J_m, J_n\} = i\delta_{j_n}J_m = \frac{L^2}{32\pi^2 G}m\delta_{m+n,0}. \tag{25c}$$

The algebra described by the relations (25) corresponds to a Virasoro-Kac-Moody $U(1)$ algebra, the symmetry algebra of a WCFT,

$$i\{L_m, L_n\} = (m-n)L_{m+n} + \frac{c}{12}m^3\delta_{m+n,0}, \tag{26a}$$

$$i\{L_m, J_n\} = -nJ_{m+n} - i\kappa m^2\delta_{m+n,0}, \tag{26b}$$

$$i\{J_m, J_n\} = \frac{k}{2}m\delta_{m+n,0}, \tag{26c}$$

with central charges $c$, $\kappa$ and $k$

$$c = 0, \quad \kappa = \frac{L}{16\pi i G}, \quad k = \frac{L^2}{16\pi^2 G}. \tag{27}$$

Note that the central extensions obtained here are manifestly real, since $L \in i\mathbb{R}$.

## 2.4 Boundary conditions in Schwarzschild-like coordinates

Previously, we applied the Godet-Marteau boundary conditions on the extremal BTZ black hole by introducing a new system of coordinates (with a retarded time $u$). With this system of coordinates, the metric was written in a form that was similar to the Bondi gauge for AdS$_2$. Here, we perform our analysis in the Schwarzschild-like system of coordinates. In these coordinates, the metric of the extremal BTZ black hole (in the NHL) reads (3). Upon rescaling $\rho \to (r_h \rho)/2$ and $\varphi \to \varphi/(2r_h)$, the metric (3) becomes

$$ds^2 = \frac{1}{4}\left(\frac{d\rho^2}{\rho^2} - \rho^2 d\tau^2\right) + \frac{1}{4}(d\varphi + \rho d\tau)^2 \, . \tag{28}$$

We now impose Godet-Marteau boundary conditions on this metric by applying a finite coordinate transformation given by

$$\tau \to \mathcal{F}(\tau), \quad \rho \to \frac{1}{\mathcal{F}'}\left(\rho + \mathcal{G}'(\tau)\right), \quad \varphi \to \varphi - \mathcal{G}(\tau). \tag{29}$$

Defining a function $\mathcal{H}(\tau) \equiv \mathcal{F}''(\tau)/\mathcal{F}'(\tau)$, this transformation yields the metric components

$$g_{\tau\tau} = -\frac{1}{2}\rho\mathcal{G}'(\tau) - \frac{1}{4}\mathcal{G}'(\tau)^2 + \frac{\left(\left(\rho + \mathcal{G}'(\tau)\right)\mathcal{H}(\tau) - \mathcal{G}''(\tau)\right)^2}{4(\rho + \mathcal{G}'(\tau))^2} \, , \tag{30a}$$

$$g_{\tau\rho} = \frac{-\left(\rho + \mathcal{G}'(\tau)\right)\mathcal{H}(\tau) + \mathcal{G}''(\tau)}{4(\rho + \mathcal{G}'(\tau))^2}, \quad g_{\tau\varphi} = \frac{\rho}{4} \, , \tag{30b}$$

$$g_{\rho\rho} = \frac{1}{4(\rho + \mathcal{G}'(\tau))^2} \, , \quad g_{\rho\varphi} = 0, \quad g_{\varphi\varphi} = \frac{1}{4}. \tag{30c}$$

The metric of the extremal BTZ black hole (in the NHL) corresponds to (30) with $\mathcal{G}'(\tau) = 0$ and $\mathcal{H}(\tau) = 0$.

The asymptotic Killing vectors generating the transformations (29) are given by

$$\xi = \epsilon(\tau)\partial_\tau - (\rho\epsilon'(\tau) + \zeta'(\tau))\partial_\rho + \zeta(\tau)\partial_\varphi \, , \tag{31}$$

where $\epsilon(\tau)$ and $\zeta(\tau)$ are two arbitrary functions of $\tau$. By applying the Lie derivative on the metric, we find the variations of $\mathcal{G}'(\tau)$ and $\mathcal{H}(\tau)$:

$$\delta_\xi \mathcal{G}'(\tau) = \epsilon'(\tau)\mathcal{G}'(\tau) + \epsilon(\tau)\mathcal{G}''(\tau) - \zeta'(\tau), \tag{32}$$

$$\delta_\xi \mathcal{H}(\tau) = \epsilon'(\tau)\mathcal{H}(\tau) + \epsilon''(\tau) + \epsilon(\tau)\mathcal{H}'(\tau). \tag{33}$$

In the following, we assume that $\tau$ is periodic with period $L$. We define modes as

$$l_n = \xi\left(\epsilon = \frac{L}{2\pi}e^{2\pi i n\tau/L}, \zeta = 0\right), \quad j_n = \xi\left(\epsilon = 0, \zeta = \frac{L}{2\pi i}e^{2\pi i n\tau/L}\right). \tag{34}$$

Under the Lie bracket, these modes satisfy the warped Witt algebra (17).

Here, the integral considered in (18) for the computation of the charges is taken over $\tau$ while $\rho \to \infty$ and $\varphi$ is constant. Explicitly computing the charges yields

$$Q_\xi = \frac{1}{32\pi G}\int_0^L d\tau\left(2\zeta(\tau)\mathcal{G}'(\tau) - \epsilon(\tau)\mathcal{G}'(\tau)^2 + 2\epsilon'(\tau)\mathcal{H}(\tau) + \epsilon(\tau)\mathcal{H}(\tau)^2\right), \tag{35}$$

where the metric of the extremal black hole in the NHL, which has $\mathcal{G}'(\tau) = \mathcal{H}(\tau) = 0$, has been chosen as the background metric. We define

$$L_n = Q_{l_n} = \frac{1}{32\pi G}\int_0^L d\tau\left(-\mathcal{G}'(\tau)^2 + \frac{4\pi i n}{L}\mathcal{H}(\tau) + \mathcal{H}(\tau)^2\right)\frac{L}{2\pi}e^{2\pi i n\tau/L} \, , \tag{36}$$

$$J_n = Q_{j_n} = \frac{1}{32\pi G}\int_0^L d\tau(2\mathcal{G}'(\tau))\frac{L}{2\pi i}e^{2\pi i n\tau/L} \, . \tag{37}$$

The algebra of the charges $L_n$ and $J_n$ is given by (26a)-(26c) with central charges

$$c^* = \frac{3}{2G}, \quad \kappa^* = 0, \quad k^* = \frac{L^2}{16\pi^2 G}, \tag{38}$$

which are different from those found in (27). We would like to know if it is possible to relate algebras (26) with different central charges, given by (27) and (38), respectively. By defining new surface charges [54]

$$L_n^* := L_n + \frac{2i\kappa}{k} n J_n, \tag{39}$$

it is possible to go from an algebra with central charges $c$, $\kappa$ and $k$ to a new algebra with central charges given by

$$c^* = c - \frac{24\kappa^2}{k}, \quad \kappa^* = 0, \quad k^* = k. \tag{40}$$

Using this relation, the central charges found here, $(c^*, \kappa^*, k^*)$, can be related to those found in (27), $(c, \kappa, k)$. Explicitly, we have

$$c^* = 0 - 24 \left( \frac{-L^2}{(16\pi)^2 G^2} \right) \frac{16\pi^2 G}{L^2} = \frac{3}{2G}, \tag{41}$$

and the relations for $\kappa^*$ and $k^*$ are trivial. Note that $c^*$ is recognized as the Brown-Henneaux central charge for $AdS_3$ gravity [14].

From a holographic perspective, the redefiniton of the charges, equation (39), corresponds to twisting the stress tensor in the boundary theory. This is the boundary counterpart of performing the change of coordinates from Eddington-Finkelstein-like coordinates to Schwarzschild-like coordinates in the bulk.

## 3 Extremal Kerr

### 3.1 Geometry and NHEK

The analysis of the previous sections can also be applied to extremal Kerr black holes. The metric of the extremal Kerr black hole in Boyer-Lindquist coordinates reads

$$ds^2 = -\frac{\Delta}{\rho^2} \left( dt - a \sin^2\theta \, d\phi \right)^2 + \frac{\sin^2\theta}{\rho^2} \left( (r^2 + a^2) d\phi - a dt \right)^2 + \frac{\rho^2}{\Delta} dr^2 + \rho^2 d\theta^2, \tag{42}$$

where

$$\Delta = (r-a)^2, \quad \rho^2 = r^2 + a^2 \cos^2\theta, \quad a = GM. \tag{43}$$

We consider the change of coordinates

$$\hat{r} = \frac{r - GM}{\lambda GM}, \quad \hat{t} = \frac{\lambda t}{2GM}, \quad \hat{\phi} = \phi - \frac{t}{2GM}, \tag{44}$$

and take the limit $\lambda \to 0$, yielding the near-horizon extremal Kerr (NHEK) geometry

$$ds^2 = G^2 M^2 (1 + \cos^2\theta) \left( \frac{d\hat{r}^2}{\hat{r}^2} + d\theta^2 - \hat{r}^2 d\hat{t}^2 \right) + \frac{4G^2 M^2 \sin^2\theta}{1 + \cos^2\theta} (d\hat{\phi} + \hat{r} d\hat{t})^2. \tag{45}$$

Hereafter, we will omit "^" of the coordinates. In order to apply Godet-Marteau boundary conditions to this metric, we write it in a system of coordinates similar to the Bondi coordinates

$$t = u - \frac{1}{r}, \quad \phi = \varphi - \ln r, \tag{46}$$

such that the metric becomes

$$ds^2 = G^2 M^2 (1 + \cos^2\theta)(-r^2 du^2 - 2du dr + d\theta^2) + \frac{4G^2 M^2 \sin^2\theta}{1 + \cos^2\theta} (d\varphi + r du)^2. \tag{47}$$

## 3.2 Phase space and asymptotic Killing vectors

Inspired by the Godet-Marteau boundary conditions for $AdS_2$ [54], we consider the following family of metrics

$$ds^2 = G^2 M^2 (1 + \cos^2\theta)((-r^2 du^2 + 2P(u)r + 2T(u))du^2 - 2dudr + d\theta^2)$$
$$+ \frac{4G^2 M^2 \sin^2\theta}{1 + \cos^2\theta}(d\varphi + rdu)^2, \tag{48}$$

where $P$ and $T$ are arbitrary functions of $u$. They can be obtained from (47) by applying a finite coordinate transformation given by

$$u \to \mathcal{F}(u), \quad r \to \frac{1}{\mathcal{F}'}\left(r + \mathcal{G}'(u)\right), \quad \varphi \to \varphi - \mathcal{G}(u), \quad \theta \to \theta. \tag{49}$$

The functions $P, T, \mathcal{F}$ and $\mathcal{G}$ are related by

$$P(u) = -\mathcal{G}'(u) + \frac{\mathcal{F}''(u)}{\mathcal{F}'(u)}, \tag{50}$$

$$T(u) = -\frac{1}{2}\mathcal{G}'(u)^2 + \mathcal{G}'(u)\frac{\mathcal{F}''(u)}{\mathcal{F}'(u)} - \mathcal{G}''(u). \tag{51}$$

The asymptotic Killing vectors generating the transformations (49) are given by

$$\xi = \epsilon(u)\partial_u - (r\epsilon'(u) + \zeta'(u))\partial_r + \zeta(u)\partial_\varphi, \tag{52}$$

where $\epsilon(u)$ and $\zeta(u)$ are two arbitrary functions of $u$. By applying the Lie derivative on the metric (48), we can also find

$$\delta_\xi P = \epsilon P' + \epsilon' P + \epsilon'' + \zeta', \tag{53}$$
$$\delta_\xi T = \epsilon T' + 2\epsilon' T - \zeta' P + \zeta''. \tag{54}$$

From now on we assume that $u$ is periodic with period $L$. We define the modes

$$l_n = \xi\left(\frac{L}{2\pi}e^{2\pi i n u/L}, 0\right), \quad j_n = \xi\left(0, \frac{L}{2\pi i}e^{2\pi i n u/L}\right), \tag{55}$$

where $n \in \mathbb{Z}$. Under the Lie bracket, these modes satisfy the warped Witt algebra (17).

## 3.3 Charge algebra

We can now compute the surface charges by using the expression (18). For this, we integrate over $u$ and $\theta$ while keeping $\varphi$ fixed and taking $r \to \infty$. Defining

$$h_{\mu\nu} \equiv \delta g_{\mu\nu} = \frac{\partial g_{\mu\nu}}{\partial P}\delta P + \frac{\partial g_{\mu\nu}}{\partial T}\delta T, \tag{56}$$

we compute

$$\delta Q_\xi = \frac{G^2 M^2}{4\pi G}\int_0^L du \int_0^\pi d\theta \sin\theta(\delta T \, \epsilon - \delta P \, \zeta), \tag{57}$$

which upon integration yields

$$Q_\xi = \frac{GM^2}{2\pi}\int_0^L du(T(u)\epsilon(u) - P(u)\zeta(u)), \tag{58}$$

where the NHEK geometry, which has $P(u) = T(u) = 0$, has been chosen as the background metric. In particular, we define

$$L_n = Q_{l_n} = \frac{GM^2}{2\pi} \int_0^L du T(u) \frac{L}{2\pi} e^{2\pi inu/L} , \tag{59}$$

$$J_n = Q_{j_n} = -\frac{GM^2}{2\pi} \int_0^L du P(u) \frac{L}{2\pi i} e^{2\pi inu/L} . \tag{60}$$

The charges $L_n$ and $J_n$ respect the algebra (26) with central charges given by

$$c = 0, \quad \kappa = \frac{LGM^2}{2\pi i}, \quad k = \frac{L^2 GM^2}{2\pi^2} . \tag{61}$$

## 3.4 Boundary conditions in Boyer-Lindquist coordinates

So far, we studied the NHEK geometry by writing it in a new system of coordinates (with a retarded time $u$). Now, we perform the same analysis in Boyer-Lindquist coordinates. Again, we obtain a phase space of metrics from (45) by applying the finite coordinate transformation

$$t \to \mathcal{F}(t), \quad r \to \frac{1}{\mathcal{F}'}\left(r + \mathcal{G}'(t)\right), \quad \phi \to \phi - \mathcal{G}(t). \tag{62}$$

Defining $\mathcal{H}(t) \equiv \mathcal{F}''(t)/\mathcal{F}'(t)$, yields

$$g_{tt} = \frac{4r^2 G^2 M^2 \sin^2\theta}{1 + \cos^2\theta} - G^2 M^2 (1 + \cos^2\theta)\left(r + \mathcal{G}'(t)\right)^2$$
$$+ \frac{G^2 M^2 (1 + \cos^2\theta)}{(r + \mathcal{G}'(t))^2}\left(\left(r + \mathcal{G}'(t)\right)\mathcal{H}(t) - \mathcal{G}''(t)\right)^2, \tag{63a}$$

$$g_{tr} = -G^2 M^2 (1 + \cos^2\theta)\frac{\left(\left(r + \mathcal{G}'(t)\right)\mathcal{H}(t) - \mathcal{G}''(t)\right)}{(r + \mathcal{G}'(t))^2} , \tag{63b}$$

$$g_{t\theta} = 0, \quad g_{t\phi} = \frac{4r G^2 M^2 \sin^2\theta}{1 + \cos^2\theta} , \tag{63c}$$

$$g_{rr} = \frac{G^2 M^2 (1 + \cos^2\theta)}{(r + \mathcal{G}'(t))^2}, \quad g_{r\theta} = 0, \quad g_{r\phi} = 0, \tag{63d}$$

$$g_{\theta\theta} = G^2 M^2 (1 + \cos^2\theta), \quad g_{\theta\phi} = 0, \quad g_{\phi\phi} = \frac{4G^2 M^2 \sin^2\theta}{1 + \cos^2\theta} , \tag{63e}$$

where the NHEK geometry is obtained by setting $\mathcal{G}'(t) = 0$ and $\mathcal{H}(t) = 0$. Hence, the order of the non-zero fluctuations of the boundary metric is given by

$$h_{tt} = \mathcal{O}(r), \quad h_{tr} = \mathcal{O}\left(r^{-1}\right), \quad h_{rr} = \mathcal{O}\left(r^{-3}\right). \tag{64}$$

The asymptotic Killing vectors generating the transformations (62) are given by

$$\xi(\epsilon, \zeta) = \epsilon(t)\partial_t + \left(-r\epsilon'(t) - \zeta'(t)\right)\partial_r + \zeta(t)\partial_\phi , \tag{65}$$

where $\epsilon(t)$ and $\zeta(t)$ are two arbitrary functions of $t$.

We recall that the group of exact isometries of the NHEK geometry, $SL(2, \mathbb{R}) \times U(1)$, is generated by the Killing vectors

$$\xi_{-1} = \partial_t, \quad \xi_0 = t\partial_t - r\partial_r, \quad \xi_1 = \left(t^2 + \frac{1}{r^2}\right)\partial_t - 2tr\partial_r - \frac{2}{r}\partial_\phi , \tag{66}$$

$$\xi_\phi = \partial_\phi . \tag{67}$$

Comparing these vectors with (65), we find that $\xi_{-1} = \xi(\epsilon = 1, \zeta = 0)$, $\xi_0 = \xi(\epsilon = t, \zeta = 0)$, $\xi_\phi = \xi(\epsilon = 0, \zeta = 1)$ and that $\xi_1$ correspond to $\xi(\epsilon = t^2, \zeta = 0)$ up to subleading terms in $r$. Hence, the asymptotic symmetry group contains all the exact isometries of the NHEK geometry, which was not the case for the boundary conditions studied in [64].

By applying the Lie derivative on (63), we find

$$\delta_\xi \mathcal{G}'(t) = \epsilon'(t)\mathcal{G}'(t) + \epsilon(t)\mathcal{G}''(t) - \zeta'(t), \tag{68}$$

$$\delta_\xi \mathcal{H}(t) = \epsilon'(t)\mathcal{H}(t) + \epsilon''(t) + \epsilon(t)\mathcal{H}'(t). \tag{69}$$

From now on we assume that $t$ is periodic with period $L$. We define modes as

$$l_n = \xi\left(\frac{L}{2\pi}e^{2\pi int/L}, 0\right), \quad j_n = \xi\left(0, \frac{L}{2\pi i}e^{2\pi int/L}\right), \tag{70}$$

with $n \in \mathbb{Z}$, which satisfy (17).

Here, the integral considered in (18) for the computation of the charges is taken over $t$ and $\theta$ while $r \to \infty$ and $\phi$ is constant. Computing the charges explicitly, we find

$$Q_\xi = \frac{GM^2}{4\pi}\int_0^L dt (2\mathcal{G}'(t)\zeta(t) - \mathcal{G}'(t)^2\epsilon(t) + 2\epsilon'(t)\mathcal{H}(t) + \epsilon(t)\mathcal{H}(t)^2), \tag{71}$$

where the NHEK geometry, which has $\mathcal{G}'(t) = \mathcal{H}(t) = 0$, has been chosen as the background metric. In particular, we define

$$L_n = Q_{l_n} = \frac{GM^2}{4\pi}\int_0^L dt\left(-\mathcal{G}'(t)^2 + \frac{4\pi in}{L}\mathcal{H}(t) + \mathcal{H}(t)^2\right)\frac{L}{2\pi}e^{2\pi int/L}, \tag{72}$$

$$J_n = Q_{j_n} = \frac{GM^2}{4\pi}\int_0^L dt\, 2\,\mathcal{G}'(t)\frac{L}{2\pi i}e^{2\pi int/L}. \tag{73}$$

The charges $L_n$ and $J_n$ fulfill the algebra (26) with central charges given by

$$c^* = 12GM^2 = 12J, \quad \kappa^* = 0, \quad k^* = \frac{L^2GM^2}{2\pi^2} = \frac{JL^2}{2\pi^2}. \tag{74}$$

The algebra (26) with central charges (74), $(c^*, \kappa^*, k^*)$ can be related to the one with central charges (61), $(c, \kappa, k)$, by the transformation (39) and (40). Indeed, we have

$$c^* = 0 - 24\left(\frac{LGM^2}{2\pi i}\right)^2\frac{2\pi^2}{L^2GM^2} = 12GM^2, \tag{75}$$

and the relations for $\kappa^*$ and $k^*$ are trivial. Here $c^*$ is recognized as the Kerr/CFT central charge [64].

## 3.5 Comparison to other boundary conditions for extremal Kerr black holes

We now compare our results with those obtained in [94]. There, the perturbations defined on the background metric (45) were

$$h_{tt} = \mathcal{O}\left(r^0\right), \quad h_{tr} = \mathcal{O}\left(r^{-3}\right), \quad h_{t\theta} = \mathcal{O}\left(r^{-3}\right), \quad h_{t\phi} = \mathcal{O}\left(r^{-2}\right), \tag{76a}$$

$$h_{rr} = \mathcal{O}\left(r^{-4}\right), \quad h_{r\theta} = \mathcal{O}\left(r^{-4}\right), \quad h_{r\phi} = \mathcal{O}\left(r^{-3}\right), \tag{76b}$$

$$h_{\theta\theta} = \mathcal{O}\left(r^{-3}\right), \quad h_{\theta\phi} = \mathcal{O}\left(r^{-3}\right), \quad h_{\phi\phi} = \mathcal{O}\left(r^{-2}\right), \tag{76c}$$

and the vectors solving the asymptotic Killing equation took the general form

$$\xi = \left(\epsilon(t) + \frac{\epsilon''(t)}{2r^2} + \mathcal{O}\left(r^{-3}\right)\right)\partial_t + \left(-r\epsilon'(t) + \frac{\epsilon'''(t)}{2r} + \mathcal{O}\left(r^{-2}\right)\right)\partial_r$$
$$+ \left(\mathcal{C} - \frac{\epsilon''(t)}{r} + \mathcal{O}\left(r^{-3}\right)\right)\partial_\phi + \mathcal{O}\left(r^{-3}\right)\partial_\theta, \tag{77}$$

where $\epsilon(t)$ is an arbitrary function of $t$ and $\mathcal{C}$ is an arbitrary constant. The boundary conditions (76) are different from ours, compare equation (64). Neglecting the subleading terms, we see that (65) reduces to (77) upon setting $\zeta(t) = \mathcal{C} = $ const. Hence, in both cases the expression (77) contains the vectors (66)-(67) generating the $SL(2,\mathbb{R}) \times U(1)$ group of isometries.

In [94] it is claimed that the charges associated to the vectors (77) with $\mathcal{C} = 0$ form a Virasoro algebra with vanishing central extension, contrary to our result. Indeed, restricting to a subset of our charges by considering only asymptotic Killing vectors (65) that have $\zeta(t) = 0$, we obtain a Virasoro algebra (26a) with non-zero central charge.

Different boundary conditions encompassing the NHEK geometry were also presented in [103, 104]. Starting from the background metric (45), a phase space of metrics was obtained by applying a finite coordinate transformation

$$t \to f(t) + \frac{2f''(t)f'(t)^2}{4r^2f'(t)^2 - f''(t)^2},$$
$$r \to \frac{4r^2f'(t)^2 - f''(t)^2}{4rf'(t)^3}, \tag{78}$$
$$\phi \to \phi + \log\left(\frac{2rf'(t) - f''(t)}{2rf'(t) + f''(t)}\right),$$

yielding the line element

$$ds^2 = G^2M^2(1 + \cos^2\theta)\left(-r^2\left(1 + \frac{\{f(t), t\}}{2r^2}\right)^2 dt^2 + \frac{dr^2}{r^2} + d\theta^2\right)$$
$$+ \frac{4G^2M^2\sin^2\theta}{1 + \cos^2\theta}\left(d\phi + r\left(1 - \frac{\{f(t), t\}}{2r^2}\right)dt\right)^2, \tag{79}$$

with the Schwarzian derivative

$$\{f(t), t\} = \left(\frac{f''}{f'}\right)' - \frac{1}{2}\left(\frac{f''}{f'}\right)^2. \tag{80}$$

Equivalently, the components of this metric read

$$g_{tt} = \frac{4r^2G^2M^2\sin^2\theta}{1 + \cos^2\theta}\left(1 - \frac{\{f(t), t\}}{2r^2}\right)^2 - G^2M^2(1 + \cos^2\theta)r^2\left(1 + \frac{\{f(t), t\}}{2r^2}\right)^2, \tag{81a}$$

$$g_{tr} = 0, \quad g_{t\theta} = 0, \quad g_{t\phi} = \frac{4G^2M^2\sin^2\theta}{1 + \cos^2\theta}r\left(1 - \frac{\{f(t), t\}}{2r^2}\right), \tag{81b}$$

$$g_{rr} = \frac{G^2M^2(1 + \cos^2\theta)}{r^2}, \quad g_{r\theta} = 0, \quad g_{r\phi} = 0, \tag{81c}$$

$$g_{\theta\theta} = G^2M^2(1 + \cos^2\theta), \quad g_{\theta\phi} = 0, \quad g_{\phi\phi} = \frac{4G^2M^2\sin^2\theta}{1 + \cos^2\theta}, \tag{81d}$$

which are different from the components (63) that we obtained from applying the transformation (62). The order of the non-zero fluctuations of the boundary metric

$$h_{tt} = \mathcal{O}\left(r^{-2}\right), \quad h_{t\varphi} = \mathcal{O}\left(r^{-1}\right), \tag{82}$$

are different from (76) and ours, compare equation (64). Furthermore, while here the components only depend on one free function of $t$, our class of metrics (63a)-(63e) depends on two. Expanding $f(t) = t + \epsilon(t) + \mathcal{O}(\epsilon^2)$, the asymptotic Killing vectors generating the transformations (78) are given by

$$\xi = \left(\epsilon(t) + \frac{\epsilon''(t)}{2r^2}\right)\partial_t - r\epsilon'(t)\partial_r - \frac{\epsilon''(t)}{r}\partial_\phi\,, \tag{83}$$

where $\epsilon(t)$ is an arbitrary function of $t$. Again, up to subleading terms, these vectors are a subset of the vectors (65), obtained by setting $\zeta(t) = 0$.

## 4 Ultra-cold Kerr-dS

### 4.1 Geometry and phase space

In this section, we study the near-horizon geometry of the Kerr-dS black hole in the ultracold limit where the inner, outer and cosmological horizon coincide. In this limit, the metric takes the form [42]

$$\frac{ds^2}{\ell^2} = \Gamma(\theta)\left(-dt^2 + dr^2 + \alpha(\theta)d\theta^2\right) + \gamma(\theta)(d\phi + \bar{k}rdt)^2\,, \tag{84}$$

with

$$\Gamma(\theta) = \frac{\sqrt{2\sqrt{3}-3}\left((3-2\sqrt{3})\cos^2(\theta)-1\right)}{2(\sqrt{3}-3)}\,, \qquad \alpha(\theta) = \frac{2\sqrt{14\sqrt{3}-24}}{(7\sqrt{3}-12)\cos^2(\theta)+\sqrt{3}}\,, \tag{85}$$

$$\gamma(\theta) = \frac{\sin^2(\theta)\left((15\sqrt{3}-26)\cos^2(\theta)+\sqrt{3}-2\right)}{3(4\sqrt{3}-7)\cos(2\theta)+8\sqrt{3}-15}\,, \qquad \bar{k} = -\sqrt{3}\,, \tag{86}$$

where the bar has been introduced to avoid possible confusions between the parameter $\bar{k}$ with the central extension $k$. Here, we have chosen our units such that the cosmological constant $\Lambda = 3/\ell^2$, with $\ell$ being the dS radius. The sign of $\bar{k}$ is arbitrary and can be changed by sending $t \to -t$. We change to Eddington-Finkelstein-like coordinates

$$u = t - r\,, \quad \phi = \bar{\varphi} - \frac{\bar{k}r^2}{2}\,, \tag{87}$$

yielding

$$\frac{ds^2}{\ell^2} = \Gamma(\theta)\left(-du^2 - 2dudr + \alpha(\theta)d\theta^2\right) + \gamma(\theta)\left(d\bar{\varphi} + \bar{k}rdu\right)^2\,. \tag{88}$$

Upon setting

$$\bar{\varphi} = \bar{k}\varphi\,, \qquad \gamma(\theta) = \frac{\bar{\gamma}(\theta)}{\bar{k}^2}\,, \tag{89}$$

we get

$$\frac{ds^2}{\ell^2} = \Gamma(\theta)(-du^2 - 2dudr + \alpha(\theta)d\theta^2) + \bar{\gamma}(\theta)(d\varphi + rdu)^2\,. \tag{90}$$

Inspired by [63], we consider the following family of metrics

$$\frac{ds^2}{\ell^2} = \Gamma(\theta)\left(2(P(u)r + T(u))du^2 - 2dudr + \alpha(\theta)d\theta^2\right) + \bar{\gamma}(\theta)(d\varphi + rdu)^2\,, \tag{91}$$

where $P$ and $T$ are arbitrary functions of $u$. This family can be obtained by applying the finite coordinate transformation

$$u \to \mathcal{F}(u), \quad r \to \frac{1}{\mathcal{F}'}\left(r + \mathcal{G}'(u)\right), \quad \varphi \to \varphi - \mathcal{G}(u), \tag{92}$$

to (90). The functions $P, T, \mathcal{F}$ and $\mathcal{G}$ are related by

$$T(u) = -\frac{1}{2}\mathcal{F}'(u)^2 - \mathcal{G}''(u) + \frac{\mathcal{G}'(u)\mathcal{F}''(u)}{\mathcal{F}'(u)}, \quad P(u) = \frac{\mathcal{F}''(u)}{\mathcal{F}'(u)}. \tag{93}$$

## 4.2 Asymptotic Killing vectors

The asymptotic Killing vectors generating the transformations (92) read

$$\xi = \epsilon(u)\partial_u - \left(r\epsilon'(u) + \zeta'(u)\right)\partial_r + \zeta(u)\partial_\varphi, \tag{94}$$

where $\epsilon(u)$ and $\zeta(u)$ are two arbitrary functions of $u$. We take the retarded time $u$ to be periodic with period $L$, and define the generators

$$l_n = \xi\left(\epsilon = \frac{L}{2\pi}e^{2\pi i n u/L}, \zeta = 0\right), \quad j_n = \xi\left(\epsilon = 0, \zeta = \frac{L}{2\pi i}e^{2\pi i n u/L}\right), \tag{95}$$

which obey (17). By applying the Lie derivative on the metric (91), we find the variations of $T(u)$ and $P(u)$

$$\delta_\xi T(u) = \left(2T(u)\epsilon'(u) + \epsilon(u)T'(u) - P(u)\zeta'(u) + \zeta''(u)\right), \tag{96a}$$

$$\delta_\xi P(u) = \left(P(u)\epsilon'(u) + \epsilon(u)P'(u) + \epsilon''(u)\right). \tag{96b}$$

## 4.3 Charge algebra

We compute the surface charges from (18), yielding

$$Q = \frac{\ell^2}{8\pi G}\int_0^L du\left(\sqrt{3}-1\right)(\epsilon(u)T(u) - \zeta(u)P(u)), \tag{97}$$

where we have integrated over a constant $r, \varphi$ surface and taken the limit $r \to \infty$. Defining

$$L_n = Q_{l_n} = \frac{L\ell^2}{16\pi^2 G}\int_0^L du\left(\sqrt{3}-1\right)e^{2\pi i n u/L}T(u), \tag{98a}$$

$$J_n = Q_{j_n} = -\frac{L\ell^2}{16\pi^2 iG}\int_0^L du\left(\sqrt{3}-1\right)e^{2\pi i n u/L}P(u), \tag{98b}$$

one readily computes that the charges $L_n, J_n$ obey (26) with $c = k = 0$ and

$$\kappa = \frac{1}{i}\frac{L\ell^2}{8\pi G}\left(\sqrt{3}-1\right). \tag{99}$$

# 5 Conclusion

In this paper, we studied new boundary conditions for the near-horizon geometries of extremal black holes in three and four dimensions. Our boundary conditions for extremal BTZ and Kerr black holes were obtained by uplifting the boundary conditions by Godet and Marteau [54],

from two to three or from two to four dimensions. In the case of the ultra-cold Kerr-dS black hole, our boundary conditions were obtained by uplifting the boundary conditions by Afshar, González, Grumiller and Vassilevich [63] from two to four dimensions. This shows that certain boundary conditions existing in 2d gravity can be uplifted to higher dimensions in a natural way.

We studied the asymptotic symmetries preserving these boundary conditions and associated charges. Our charges are computed by integrating over time on a constant azimuthal angle surface, instead of doing it vice versa (integrating over the azimuthal angle on a constant time surface). To obtain finite charges, Lorentzian time necessarily needs to be periodic — this is to be understood as the Wick rotation of the periodicity in Euclidean time. Switching angular and temporal circle for the computation of the charges is motivated from modular invariance of the putative dual field theory, as detailed in the introduction. This introduces a time scale $L = i\beta$ in our charges and central extensions, where $\beta$ is the temperature. In this way, we obtain non-trivial charges which span a Virasoro-Kac-Moody algebra, the symmetry algebra of a warped conformal field theory. The results for the central extensions are summarized in table 1. For the case of the extremal BTZ and Kerr black holes we studied boundary conditions and the associated asymptotic symmetry algebras in two different systems of coordinates. A priori, such boundary conditions are not equivalent, as when it comes to asymptotic symmetries and charges, diffeomorphisms can have non-vanishing associated charges and thus carry non-trivial information. Having different boundary conditions available, the choice of boundary conditions is related to how the asymptotic boundary of spacetime is approached — in our case: following spacelike curves in Schwarzschild-like coordinates or null curves in Eddington-Finkelstein-like coordinates. However, even if one fixes the direction of approach to the asymptotic boundary, different boundary conditions are possible. The particular choice of boundary conditions is not unique and depends on the physical situation at hand. Our analysis for extremal BTZ and Kerr black holes yields, in each case a Virasoro-Kac-Moody algebra, albeit with different central extensions. We then showed that in both cases these different central extensions can be related by a mere redefinition of generators, showing that the two algebras are isomorphic. This mirrors the analysis of [23], which studied the asymptotic symmetries of three-dimensional asymptotically AdS spacetime in Bondi gauge, yielding a Virasoro algebra, the algebra found in the previous analysis performed in the Fefferman-Graham gauge [14]. In the case of extremal Kerr black holes we relate our results to boundary conditions which have previously been studied in the literature [94, 103, 104]. For this, we have to truncate our asymptotic symmetries to a Virasoro ⊕ $u(1)$ algebra. Contrary to [94], we find, that the Virasoro algebra has non-vanishing central charge, paving the way for possible microstate countings using asymptotic symmetries, in the spirit of [64].
Lastly, we studied the near-horizon geometry of the ultra-cold Kerr-dS black hole whose holographic interpretation has so far been elusive. We find, using an uplift of the boundary conditions [63], that the asymptotic symmetries span a Virasoro-Kac-Moody algebra, thereby providing first evidence that warped conformal field theories could be the holographic dual for such black holes.

While our analysis is purely classical, our results suggest that warped conformal field theories provide a holographic description of extremal black holes. This kinematical observation, based on symmetries, could be pushed in various directions, such as entropy matchings and perturbation theory to put the proposal on firmer grounds. In line with this, the question arises, whether the quantum theories obtained by performing standard canonical quantization are unitary. In the case of the extremal Kerr and BTZ black holes, due to the isomorphism mentioned above, it suffices to consider whether representation with central charges $(c, 0, k)$ can be unitary. We answer this question in the negative, as in our case $c > 0$ and $k < 0$, c.f. table 1, due to $L \in i\mathbb{R}$ and $M \in \mathbb{R}$. However, to have unitary highest-weight representation it is

Table 1: Central charges obtained for different black holes in different systems of co-ordinates: the central charges, $(c, \kappa, k)$, found by studying the asymptotic symmetries in Eddington-Finkelstein coordinates are related to the central charges, $(c^*, \kappa^*, k^*)$, found by studying the asymptotic symmetries in Schwarzschild-like coordinates due to the isomorphism (39).

| Black hole | $c$ | $\kappa$ | $k$ | Relation between the central charges |
|---|---|---|---|---|
| Extremal BTZ $(u, r, \varphi)$ | 0 | $\frac{L}{16\pi i G}$ | $\frac{L^2}{16\pi^2 G}$ | $c^* = c - 24\kappa^2/k$, $\kappa^* = 0$, $k^* = k$ |
| Extremal BTZ $(\tau, \rho, \varphi)$ | $\frac{3}{2G}$ | 0 | $\frac{L^2}{16\pi^2 G}$ | |
| Extremal Kerr $(u, r, \theta, \varphi)$ | 0 | $\frac{LGM^2}{2\pi i}$ | $\frac{L^2 GM^2}{2\pi^2}$ | $c^* = c - 24\kappa^2/k$, $\kappa^* = 0$, $k^* = k$ |
| Extremal Kerr $(t, r, \theta, \phi)$ | $12GM^2$ | 0 | $\frac{L^2 GM^2}{2\pi^2}$ | |
| Ultra-cold Kerr-dS $(u, r, \theta, \varphi)$ | 0 | $\frac{L\ell^2(\sqrt{3}-1)}{8\pi i G}$ | 0 | |

necessary to have $c > 0$ and $k > 0$, see [56, Section 2.3]. For the case of the ultracold Kerr-dS black hole, the answer is not clear, since we cannot make the redefintion and representations with $k = 0$ and $\kappa \neq 0$ have not been discussed in the literature to the best of our knowledge.

In the context of holography, WCFTs with a positive central charge but a negative $U(1)$ level have appeared generically. Despite featuring negative norm descendant states that violate unitarity, some of their properties are kept under good control. For instance, it was shown that the modular bootstrap remains feasible in theories with mild violations of unitarity, where the negative norm states can be resummed into a Virasoro-Kac-Moody character whose contribution to the bootstrap equations is positive [60]. In fact, any WCFT with a negative level must feature at least two states with imaginary U(1) charge, rendering the Hamiltonian non-hermitian. However, that feature is essential for the WCFT counterpart of the Cardy formula to be able to reproduce the entropy of WAdS$_3$ black holes [56]. Furthermore, the study of the extremal limits of WAdS$_3$ black holes and WCFTs (in the spirit of [105] for 2d CFTs) has revealed the emergence of a universal Schwarzian sector (as expected on general grounds for extremal black holes [44, 106]), but only when the seed theory was non-unitary [62, 89].

We leave it to future work to exploit our results to establish a potential dual holographic description of extremal black holes in terms of a warped conformal field theory and study the microscopic description of extremal black holes.

# Acknowledgments

The authors thank Dionysios Anninos, Alejandra Castro, Daniel Grumiller, Tom Hartman, and Chiara Toldo for useful discussions and exchanges on the topics covered in this work. We thank Katharina Schäfer for initial collaboration on Sect. 4 of this paper.

**Funding information** RW acknowledges support of the Fonds de la Recherche Scientifique F.R.S.-FNRS (Belgium) through the PDR/OL C62/5 project "Black hole horizons: away from conformality" (2022-2025) and RW thanks the Erwin Schrödinger Institute for hospitality,

where part of this work was carried out. RW also acknowledges support by the Heising-Simons Foundation under the "Observational Signatures of Quantum Gravity" collaboration grant 2021-2818 and the U.S. Department of Energy, Office of High Energy Physics, under Award No. DE-SC0019470 during the final stages of this work. TS is a Research Fellow of the Fonds de la Recherche Scientifique F.R.S.- FNRS (Belgium). SD is a Senior Research Associate of the Fonds de la Recherche Scientifique F.R.S.-FNRS (Belgium). SD was supported in part by IISN - Belgium (convention 4.4503.15) and benefited from the support of the Solvay Family. SD acknowledges support of the Fonds de la Recherche Scientifique F.R.S.- FNRS (Belgium) through the CDR project C 60/5 - CDR/OL "Horizon holography: black holes and field theories" (2020-2022), and the PDR/OL C62/5 project "Black hole horizons: away from conformality" (2022-2025).

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
