# Peer review of "Boundary Conditions for Extremal Black Holes from 2d Gravity"

_SciPost Physics, doi:SciPost Phys. 16, 141 (2024)_

## Round 1 · Referee Report · Anonymous (Referee 1) · 2024-4-6

Strengths
Weaknesses
Report
Requested changes
1-In order to perform the mode expansion, the authors assume that $\tau$ in eq(34) is periodic. The authors should explain why such an identification is reasonable in a Lorentzian geometry, and what the implication in the putative dual theory. Relatedly, the central charges in eq.(25) depend on the Length $L$ which is arbitrarily chosen. The authors should explain why such central charges are physical. 2-The authors should provide more explanations on boundary conditions in different coordinates. Are these boundary conditions equivalent in the sense that they define the same quantum theory? If these boundary conditions are not equivalent, is there a preferred one? What is the interpretation of the relation eq.(39) that relates charges in different coordinates? How do we understand these from the perspective of the holographic dual? 3- What is the interpretation of imaginary level $\kappa$ such as in eq.(61)? Is the resulting dual theory non-unitary? If so, how do we make sense of it ?
We thank the referee for the careful reading of the manuscript and for their comments and questions. We have supplemented our paper with a conclusion to address their comments.
- We have addressed the assumption of Lorentzian time being periodic in the conclusion (paragraph 2), at the end of page 6 and on page 7 under equation (19). This is in addition to the existing explanation in the introduction (page 4 of the paper, paragraph 1). The periodicity of Lorentzian time in our paper should be understood as the Wick rotation of the periodicity in Euclidean time. We motivate switching angular and temporal cycle for the computation of the charges by assuming modular invariance of the putative dual quantum field theory. This introduces the length scale $L = I \beta$ in our charges, where $\beta$ is the temperature, hence it is not arbitrary. Our central charges are physical in the sense that these are central charges in general relativity that we compute with standard covariant phase space methods; however, the switching of the cycles requires the assumption that the putative dual theory is modular invariant, which is the case for warped conformal field theories or conformal field theories, but does not hold in general. In addition, by endowing the time coordinate with a complex period, we complexify our manifold.
- We have addressed this point in the conclusion (paragraph 2) and at the very end of section 2.4. In particular, relation (39) shows that the two algebras can be related by a mere redefinition of generators and are thus isomorphic. Hence, these boundary conditions are equivalent in the sense that they give rise to the same asymptotic symmetry algebra (up to isomorphisms). Relation (39), which, from a holographic perspective, corresponds to twisting the stress tensor in the boundary theory, can be understood to arise from the change of coordinates in the bulk from Schwarzschild-like to Eddington-Finkelstein-like coordinates.
- The central extension $\kappa$ in equation (61) is manifestly real, since $L$ is imaginary. In the case of the extremal BTZ and Kerr black holes, we can use the change of generators, eq. (39), to bring the algebra into a form where unitarity has been addressed in the literature. For the extremal Kerr and BTZ black hole, we find $c > 0$ and $k < 0$. For these values, unitary highest-weight representations do not exist. We comment more on the implications of this result for holography in the conclusion (second to last paragraph). In the case of the ultracold Kerr-dS black hole only the central extension $\kappa$ is nonvanishing and we cannot use the relation (39). For this case, we do not know whether unitary representations exist (see last paragraph of conclusion).

Author: Raphaela Wutte on 2024-05-07 [id 4475]
(in reply to Report 2 on 2024-04-09)We thank the referee for the careful reading of the manuscript and their comments. We have supplemented our paper with a conclusion to address their comments.

---

## Round 1 · Referee Report · Anonymous (Referee 2) · 2024-4-9

Strengths
-
Connection between two and higher dimensional asymptotic symmetries.
-
Novel asymptotic symmetries for Kerr and Kerr-de Sitter spacetimes.
Weaknesses
-
Physical meaning of results not entirely clear.
-
Presentation is not very clear.
Report
Requested changes
-
Provide a synthesis and summary of results, compared and contrasted to previous cases.
-
Make note of cases when central charges have non-standard reality properties, or are vanishing.
-
Provide some outlook on any uniqueness of lack thereof of the new boundary conditions.

---

## Round 2 · Referee Report · Anonymous (Referee 2) · 2024-5-7

Report

I am happy with the changes made by the authors and I support the paper for publication.

Recommendation

Publish (meets expectations and criteria for this Journal)

---

## Round 2 · Referee Report · Anonymous (Referee 1) · 2024-5-8

Report

The authors have addressed all my comments. I recommend that this paper be accepted for publication on SciPost.

Recommendation

Publish (easily meets expectations and criteria for this Journal; among top 50%)

---

## Round 2 · Author Response

We have addressed the comments and questions of the referees by supplementing our paper with a conclusion. In addition, we have added several sentences in other parts of the paper for additional clarification.

---

## Round 2 · List of Changes

-) changed last sentence in first paragraph of page 4
-) changed last sentence in second paragraph of page 4
-) changed sentence directly before equation (16) on page 6
-) added 2 sentences directly below equation (16) and on page 6
-) added 2 sentences at the end of the paragraph below equation (19) on page 7
-) added one sentence below equation (27) on page 8
-) added last paragraph at the very end of section 2.4.
-) added conclusion and table

---

## Editorial Decision

published